# Experimental Control of a Methanol Catalytic Membrane Reformer †

**Alejandro Cifuentes [1,2], Maria Serra [3], Ricardo Torres [2,\*] and Jordi Llorca [1,\***

[1]    Institute of Energy Technologies, Department of Chemical Engineering and Barcelona Research Center in Multiscale Science and Engineering, Universitat Politècnica de Catalunya, EEBE, Eduard Maristany 10-14, 08019 Barcelona, Spain

[2]    Department of Fluid Mechanics, Universitat Politècnica de Catalunya, EEBE, Eduard Maristany 10-14, 08019 Barcelona, Spain

[3]    Institut de Robòtica i Informàtica Industrial, CSIC—UPC, Llorens i Artigas 4-6, 08028 Barcelona, Spain

\*    Correspondence: ricardo.torres@upc.edu (R.T.); jordi.llorca@upc.edu (J.L.)

†    This work is the follow-up of a study presented at the VIII HYCELTEC conference, 2022.

**Abstract:** A simple proportional integral (PI) controller with scheduled gain has been developed and implemented in a catalytic membrane reactor (CMR) to obtain pure hydrogen from a methanol steam reforming process. The controller is designed to track the setpoint of the pure hydrogen flow rate in the permeate side of the CMR via the manipulation of the fuel inlet flow rate. Therefore, the controller actuator is the liquid pump that provides the mixture of methanol and water to the reactor. Within the CMR, the catalytic pellets of $PdZn/ZnAl_2O_4/Al_2O_3$ have been used to facilitate the methanol steam-reforming reaction under stoichiometric conditions (S/C = 1), and Pd–Ag metallic membranes have been employed to simultaneously separate the generated hydrogen. The PI controller design is based on a mathematical model constructed using transfer functions acquired from dynamic experiments conducted with the CMR. The controller has been successfully implemented, and experimental validation tests have been carried out at 450 °C and relative pressures of 6, 8, 10, and 12 bar.

**Keywords:** hydrogen; control; dynamic modelling; methanol steam reforming; catalytic membrane reactor





## 1. Introduction

On-site and on-demand hydrogen production from liquid fuels is a viable option for feeding fuel cells for portable and transport applications, thereby circumventing the issues associated with hydrogen storage. One approach to on-site hydrogen production involves the use of fuel reformers equipped with catalytic membrane reactors (CMR). These reactors offer the advantage of combining hydrogen production and separation into a single step, significantly reducing the overall unit volume, allowing for increased hydrogen production due to the shift effect, and simplifying the process compared to a sequence of purification reactors [1]. Among the various liquid fuels suitable for hydrogen production, methanol stands out as a perfect choice due to its ready availability and low reforming temperature resulting from the absence of C-C bonds [2]. The methanol steam reforming process comprises three primary reactions:

$$CH_3OH_{(g)} + H_2O_{(g)} \leftrightharpoons CO_{2(g)} + 3H_{2(g)} \ \Delta H^{\circ}_{298} = +49.4 \text{ kJ mol}^{-1} \tag{1}$$

$$CH_3OH_{(g)} \leftrightharpoons CO_{(g)} + 2H_{2(g)} \ \Delta H^{\circ}_{298} = +92 \text{ kJ mol}^{-1} \tag{2}$$

$$CO_{(g)} + H_2O_{(g)} \leftrightharpoons CO_{2(g)} + H_{2(g)} \ \Delta H^{\circ}_{298} = -41.1 \text{ kJ mol}^{-1} \tag{3}$$

Equation (1) represents the direct methanol steam reforming reaction (MSR). Equation (2) corresponds to the methanol decomposition reaction, and Equation (3) describes the water–gas shift reaction (WGS).

The methanol steam reforming process entails the need for a heating power source to maintain the reforming temperature and a liquid pump that provides the feed load. In a CMR, hydrogen is catalytically produced during the reforming step and permeates through a palladium-based membrane, selectively allowing only hydrogen to pass through, resulting in the generation of a pure hydrogen stream [3–10]. To maintain control over the hydrogen flow rate and adapt to setpoint changes in practical applications, the use of a controller becomes essential. Several studies related to the dynamic modelling of CMRs and the application of proportional integral (PI) controllers have been documented in the literature. Hedayati et al. [11] developed a dynamic model of an ethanol-reforming system within a catalytic membrane reactor, successfully validating it under different reactor pressure conditions and feed loads. Pravin et al. conducted mathematical modelling and analysis of an integrated system comprising a reformer, a separation membrane, and a fuel cell [12], subsequently extending their research to predictive control following the addition of an auxiliary electric battery [13]. Marquez-Ruiz et al. focused on the control of a catalytic membrane reactor for the steam reforming of methane [14], developing a nonlinear model of the system and testing various control strategies. Chen et al. [15] constructed a dynamic model of a methane reformer coupled with a proton exchange membrane fuel cell (PEMFC), and they designed and applied a genetic optimization algorithm to derive multiple PI controllers for the integrated system. Kyriakides et al. implemented a mathematical model for simulating and controlling a membrane reactor used for catalytic methane reforming and implemented an integrated process [16,17]. Serra et al. designed a nonlinear predictive control for an ethanol reforming system in a catalytic membrane reactor based on a lumped parameters dynamic model [18]. Koch et al. developed a mathematical model for an ethanol CMR and implemented adaptive and predictive control on a real-time system to take account of nonlinear behaviour [19]. Stamps and Gatzke [20] presented dynamic models of a methanol-reforming reactor integrated with a PEMFC. El-Sharkh et al. [21] developed a dynamic model using transfer functions with the aim of controlling the electrical current of the fuel cell based on raw material inputs. Similarly, Wu and Pai [22] employed fuzzy control in a methanol processing unit connected to a fuel cell, while Ipsakis et al. [23] implemented a PI controller for a methanol-reforming system coupled with a fuel cell.

While comprehensive models have been developed for methanol fuel reformers equipped with CMRs, there is a notable lack of studies addressing controllability issues from an experimental standpoint. This work is dedicated to the design and implementation of a simple PI controller with scheduled gain, aimed at regulating the production of pure hydrogen (permeate flow) within a catalytic membrane reactor (CMR) during the methanol steam reforming process. The reactor utilizes a catalyst based on PdZn alloy nanoparticles anchored over $ZnAl_2O_4$ spinel supported on $Al_2O_3$ pellets. This catalyst was previously developed, characterized and tested in a prior study [24]. The fine-tuning of the parameters for the PI controller transfer function was performed through a programmed gain strategy and subsequently implemented within the CMR. The PI controller functions by controlling the high-pressure liquid pump within the system, responsible for injecting the mixture of methanol and water into the CMR. This work is the follow-up of an extended abstract presented at the VIII HYCELTEC conference, 2022 [25].

## 2. Materials and Methods

### 2.1. Experimental Setup

The schematic of the plant is depicted in Figure 1. The plant comprises a tank containing a mixture of methanol and water, which is connected to a liquid pump (Knauer® Smartline, Berlin, Germany). This pump feeds the catalytic membrane reactor. A customized Inconel membrane reactor from REB Research & Consulting® (Oak Park, MI, USA)

was used as the CMR. The reactor is equipped with four membranes, each measuring 3 inches in height and having dead-end tubes with a diameter of 1/8 inch. These tubes are coated with a 30 μm thick Pd–Ag active layer, covering a total area of 30.4 cm$^2$, and they are supported on a porous stainless-steel base. For detailed physical characteristics of the membranes, please refer to [19]. The membranes were surrounded by pellets of PdZn/ZnAl$_2$O$_4$/Al$_2$O$_3$ catalysts of 2.5 mm diameter for the methanol steam reforming [24]. The reactor is enclosed by an electrical resistance element that is controlled with an electronic controller (Fuji$^®$ PXR4, Aichi, Japan). This electrical resistance preheats the liquid before it enters the reactor and maintains the desired temperature inside the reactor. The reactor has two outlets, the membrane outlet (permeate), which contains high-purity hydrogen that permeates through the Pd–Ag metal membranes, and a residual gas outlet (retentate), through which the non-permeated hydrogen and the other gases, residual or produced during the reaction, such as CO, CO$_2$, CH$_4$, and water vapour, are released. The permeated hydrogen passes through a particulate filter and a digital mass flow sensor (Bronkhorst$^®$, Ruurlo, The Netherlands). The residual gas passes through a coil and a liquid trap to remove any unreacted reactants (methanol and water) if present. The gaseous products then pass through a coalescence filter to reduce their relative humidity. Finally, there is a digital pressure sensor and control system (Bronkhorst$^®$). The plant incorporates several thermocouples to accurately monitor the temperature throughout the process. All sensors and actuators are connected to a computer and are monitored and controlled using commercial LabVIEW$^®$ software (National Instruments, Austin, TX, USA), with a sampling time of 1 s. Both streams, permeate and retentate, undergo analysis via online gas chromatography (Agilent$^®$ 3000 A MicroGC, Santa Clara, CA, USA) using MS 5 Å, Plot U and Stabilwax columns [24].

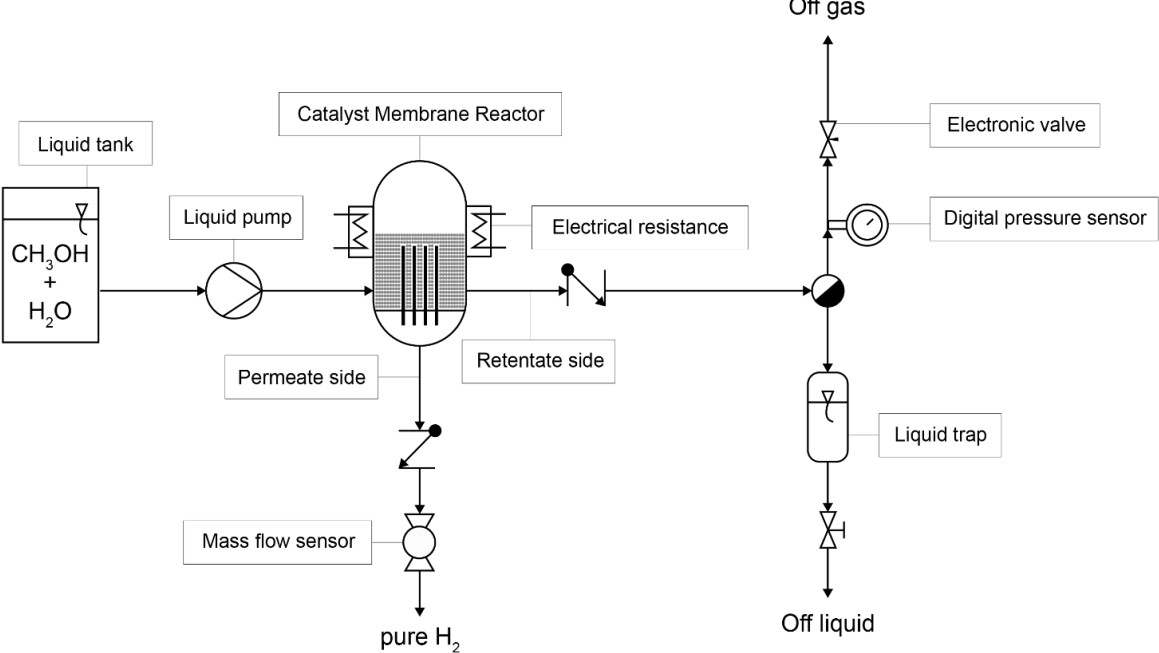

**Figure 1.** Schematic representation of the methanol reforming plant.

### 2.2. Control Variables

There are three possible input/control variables that can modify the permeated hydrogen flow rate: (i) the mass flow of liquid provided by the liquid inlet pump, (ii) the digital pressure regulator, which affects the reactor pressure, and (iii) the electrical resistance that provides heat to the reactor, affecting the reactor temperature. For a fast response of the catalytic membrane methanol reformer, temperature is not a suitable actuator, as the changes are too slow. On the other hand, pressure changes are faster, but membrane

separation efficiency is strongly affected by pressure (according to Sieverts' law), making high pressure necessary to achieve high separation efficiency of the membrane, meaning the rate of permeated hydrogen per total hydrogen produced. Modifying the pressure to lower values is detrimental to pure hydrogen recovery, as the produced hydrogen flows to the retentate side instead of the permeate. In contrast, modifying the liquid inlet flow rate is fast and does not have a significant impact on the separation efficiency of the membrane, as verified previously [24]. Consequently, the liquid pump was selected as the control variable of the PI controller.

Dynamic tests were conducted to gather information about the behaviour of the system at varying inlet flow rates and operating pressures, as documented in Table 1. The temperature was always maintained at 450 °C, a temperature that strikes a favourable balance between catalytic performance and membrane operation, as indicated in reference [24]. In all experiments, a liquid mixture of methanol and water with a stoichiometric steam-to-carbon ratio of 1 (S/C = 1, Equation (1)) was utilized. These tests involved applying step-type inputs in the inlet liquid flow (methanol and water mixture) at different pressure ranges. For each pressure level, a minimum of three series of inlet liquid tests were conducted, maintaining the pressure constant, to ensure that the system exhibited no memory effect. The results obtained were consistent across all cases.

**Table 1.** Operation conditions used for the dynamic CMR experiments.

| Reactor temperature (°C) | 450 |
|---|---|
| Reactor relative pressure (bar) | 6, 8, 10, 12 |
| Inlet flow rate ($\mu L_{liq}$/min) | 50–200 |

## 3. Results and Discussion

### 3.1. Dynamic Behaviour of the System

As an illustrative example, the experimental curves depicting the pure $H_2$ flow rate in response to step changes in the inlet liquid flow rate at 12 bars are presented in Figure 2. The blue lines represent the measured hydrogen permeated flow rates, while the orange lines represent the setpoint signal of the liquid pump. It is worth noting that complete methanol conversion was consistently attained in all instances.

Symmetric oscillations were noted in the permeated hydrogen output, which can be attributed to the characteristics of the liquid pump employed (utilizing a piston mechanism resulting in pulsatile flow rates). These experiments were replicated at the other pressures specified in Table 1. It was observed that the system exhibited faster dynamics as the pressure increased. In all instances, the system displayed a first-order response.

### 3.2. Controller Design

The strategy outlined in Figure 3 served as the basis for designing the controller. The experimental results obtained in Section 3.1. were used to derive a set of transfer functions using the commercial software MATLAB® (MathWorks, Natick, MA, USA), which, in turn, were used to construct a piecewise computational model. This model effectively captured the non-linearities of the system while maintaining efficient computational performance. Subsequently, it was implemented as a simulation model within the commercial software SIMULINK® (MathWorks, Natick, MA, USA). Building upon the computational model, an investigation into the most suitable type of PI controller for system control was conducted. Given that the system's behaviour exhibited significant variations depending on the range of the inlet flow rate, the decision was made to employ a PI controller utilizing a scheduled gain strategy. This approach allows for the adaptation of controller parameters depending on the operating region. Finally, the performance of the controlled system was validated.

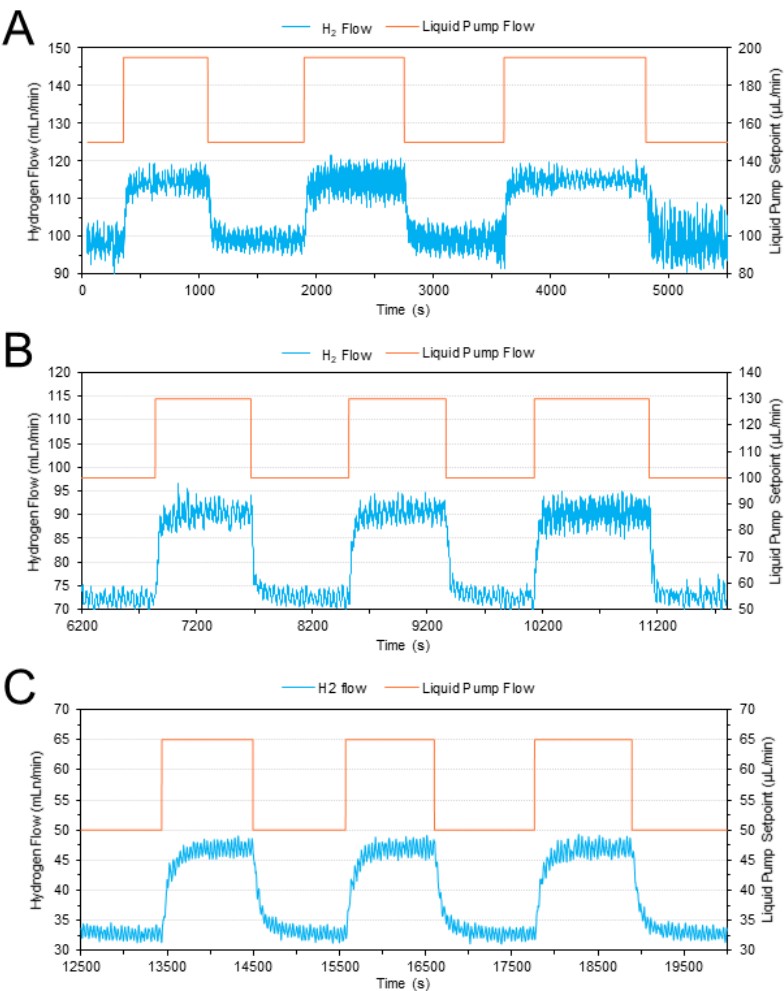

**Figure 2.** Pure hydrogen flow rate recorded against step changes in the inlet liquid flow rate at 12 bars. Inlet flow steps of 150–195 $\mu L_{liq}$/min (**A**), 100–130 $\mu L_{liq}$/min (**B**), and 50–65 $\mu L_{liq}$/min (**C**).



**Figure 3.** Scheme of the controller design strategy.

### 3.2.1. Transfer Functions

Catalytic membrane fuel reformers are inherently non-linear systems because the metallic membrane in the reactor reaches a point where it can no longer separate additional hydrogen after a specific inlet liquid flow. This phenomenon is known as membrane saturation and is a result of limitations in the mass transfer process. Following the scheduled methodology, three distinct transfer functions were derived for three different variations in the inlet flow while maintaining a constant pressure and temperature. Three specific regions of interest were defined: the low-flow zone (50–65 $\mu L_{liq}$/min), the medium-flow zone (100–130 $\mu L_{liq}$/min), and the high-flow zone (150–195 $\mu L_{liq}$/min). It is important to note that for flow rates exceeding 200 $\mu L_{liq}$/min, the membrane is unable to further separate hydrogen due to restrictions related to available surface area. The resulting transfer functions appropriately model the system as a first-order system with a time delay (as described in Equation (4)). The parameters of these transfer functions are summarized in

Table 2. Additionally, linear interpolation of the transfer function parameters was applied for flow rates between 65 and 100 $\mu L_{liq}$/min and from 130 to 150 $\mu L_{liq}$/min.

$$G(s) = \frac{k_p}{\tau_p \cdot s + 1} \cdot e^{-T_s \cdot s} \tag{4}$$

**Table 2.** Obtained parameters of the modelled transfer functions corresponding to different liquid flow inlet and pressure values.

| Pressure (Bar) | Liquid Flow Inlet ($\mu L_{liq}$/min) | Transfer Function Parameters | | |
| --- | --- | --- | --- | --- |
| | | $k_p$ | $\tau_p$ | $T_s$ |
| | 150–195 | 0.34 | 30 | 9 |
| 12 | 100–130 | 0.60 | 29 | 13 |
| | 50–65 | 0.95 | 91 | 9 |
| | 150–195 | 0.29 | 26 | 7 |
| 10 | 100–130 | 0.52 | 29 | 10 |
| | 50–65 | 0.92 | 81 | 13 |
| | 150–195 | 0.23 | 24 | 6 |
| 8 | 100–130 | 0.45 | 28 | 8 |
| | 50–65 | 0.83 | 71 | 19 |
| | 150–195 | 0.15 | 18 | 7 |
| 6 | 100–130 | 0.32 | 21 | 10 |
| | 50–65 | 0.76 | 68 | 11 |

When the model is compared with the experimental data, it is seen that the accuracy of the model oscillates between 85 and 92%. As a representative example, the comparison between the simulated response and the experimental data at 12 bar is displayed in Figure 4.

### 3.2.2. PI Controller Design

After establishing and verifying the transfer function-based model, a mathematical model of the controller was constructed using SIMULINK®. For each transfer function, a PI controller with distinct parameters was derived using the Ziegler–Nichols method (per Equation (5)). This approach facilitated the identification of optimal PI control parameters for each operational region. Subsequently, a gain-scheduled PI control was implemented for the three operating flow ranges (low, medium, and high).

$$PI\ control = K_c \left( 1 + \tau_i \frac{1}{s} \right) \tag{5}$$

The $K_c$ and $\tau_i$ parameters for the gain-scheduled controllers are summarized in Tables 3 and 4, covering various operating pressure and inlet flow values. For the sake of simplicity and due to the consistent results across the considered pressures, a decision was made to utilize average values of the $K_c$ and $\tau_i$ parameters for all pressure values.

The simulations conducted with the gain-scheduled PI controller at 12 bars, across different flow ranges, are shown in Figure 5. In Figure 5A, a slight overshoot in the controlled variable is observed following the setpoint change, but it is rapidly corrected, and the system remains stable. In Figure 5B, a minor oscillation occurs after the setpoint change, but the system reaches a steady state after a few oscillations. The behaviour of the system in Figure 5C closely resembles that of Figure 5A, with only a slight overshoot in the controlled variable.

**Table 3.** Optimal $K_c$ values of the PI controller modelled at different pressures and inlet flows.

| Pressure (Bar) | $K_c$ Low Flow (50–60 μL$_{liq}$/min) | $K_c$ Medium Flow (100–130 μL$_{liq}$/min) | $K_c$ High Flow (150–195 μL$_{liq}$/min) |
|---|---|---|---|
| 6 | 3.87 | 3.19 | 2.82 |
| 8 | 2.54 | 3.96 | 2.64 |
| 10 | 3.52 | 3.56 | 2.75 |
| 12 | 3.73 | 2.17 | 2.76 |
| Average values | 3.42 | 3.22 | 2.74 |

**Table 4.** Optimal $\tau_i$ values of the PI controller modelled at different pressures and inlet flows.

| Pressure (Bar) | $\tau_i$ Low Flow (50–60 μL$_{liq}$/min) | $\tau_i$ Medium Flow (100–130 μL$_{liq}$/min) | $\tau_i$ High Flow (150–195 μL$_{liq}$/min) |
|---|---|---|---|
| 6 | 0.015 | 0.048 | 0.085 |
| 8 | 0.015 | 0.037 | 0.055 |
| 10 | 0.012 | 0.027 | 0.047 |
| 12 | 0.013 | 0.032 | 0.036 |
| Average values | 0.014 | 0.034 | 0.051 |

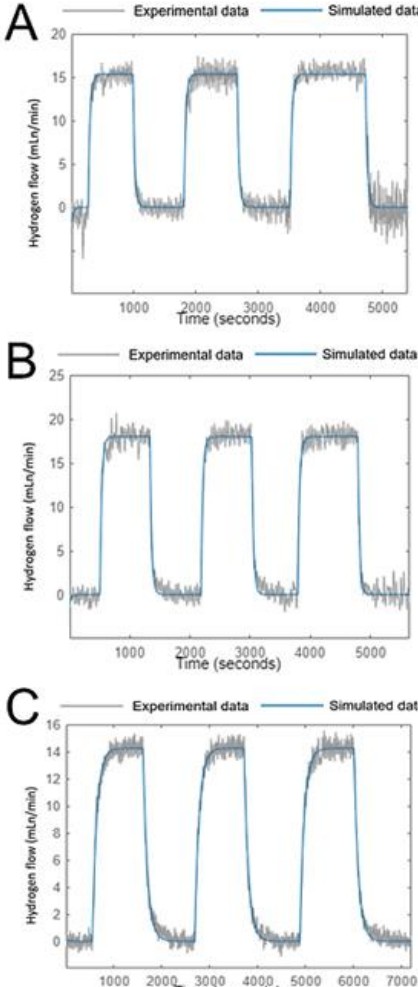

**Figure 4.** Simulated response and experimental data at 12 bars. Inlet flow rate of 150–195 μL$_{liq}$/min (**A**), 100–130 μL$_{liq}$/min (**B**), and 50–60 μL$_{liq}$/min (**C**).

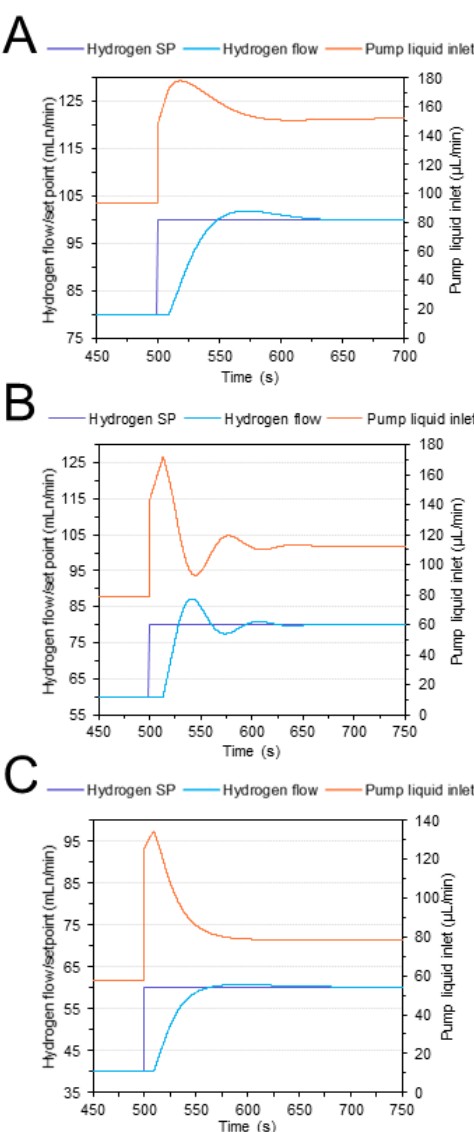

**Figure 5.** Simulated system with the PI controller at 12 bars. $H_2$ setpoint step of 80 to 100 mL/min (**A**), 60 to 80 mL/min (**B**), and 40 to 60 mL/min (**C**).

### 3.3. Implementation and Experimental Validation of the Controller

The commercial software LabVIEW$^{®}$ was employed to implement the gain-scheduled PI controller in the methanol-reforming plant. Figure 6 shows the tests conducted in the experimental system using the gain-scheduled PI controller. As previously mentioned, the setpoint was the permeated hydrogen flow rate, and the controller operated on the liquid pump to adjust the inlet liquid flow to maintain alignment with the setpoint.

The control tests conducted at 6 bars are presented in Figure 6A. The controller effectively maintains the setpoint values for all flow ranges, except for 60 $H_2$ mL/min. At this particular point, the liquid pump exhibited instability, resulting in high liquid pulses that introduced disturbances into the hydrogen output. Nevertheless, the robustness of the controller is evident as it successfully maintains the controlled variable near the setpoint, even in the presence of these disturbances. For the medium and high-pressure ranges (8–12 bar, Figure 6B–D), which are particularly conducive to hydrogen permeation through the separation membranes in the CMR, the permeated hydrogen consistently remains close to the setpoint values, exhibiting minor oscillations. In general, a small ripple of approximately ±5 $H_2$ mL/min around the setpoint value was observed. Additionally, a

slight overshoot was noted when the input liquid flow rate underwent significant changes, but this overshoot was rapidly mitigated within a few seconds.

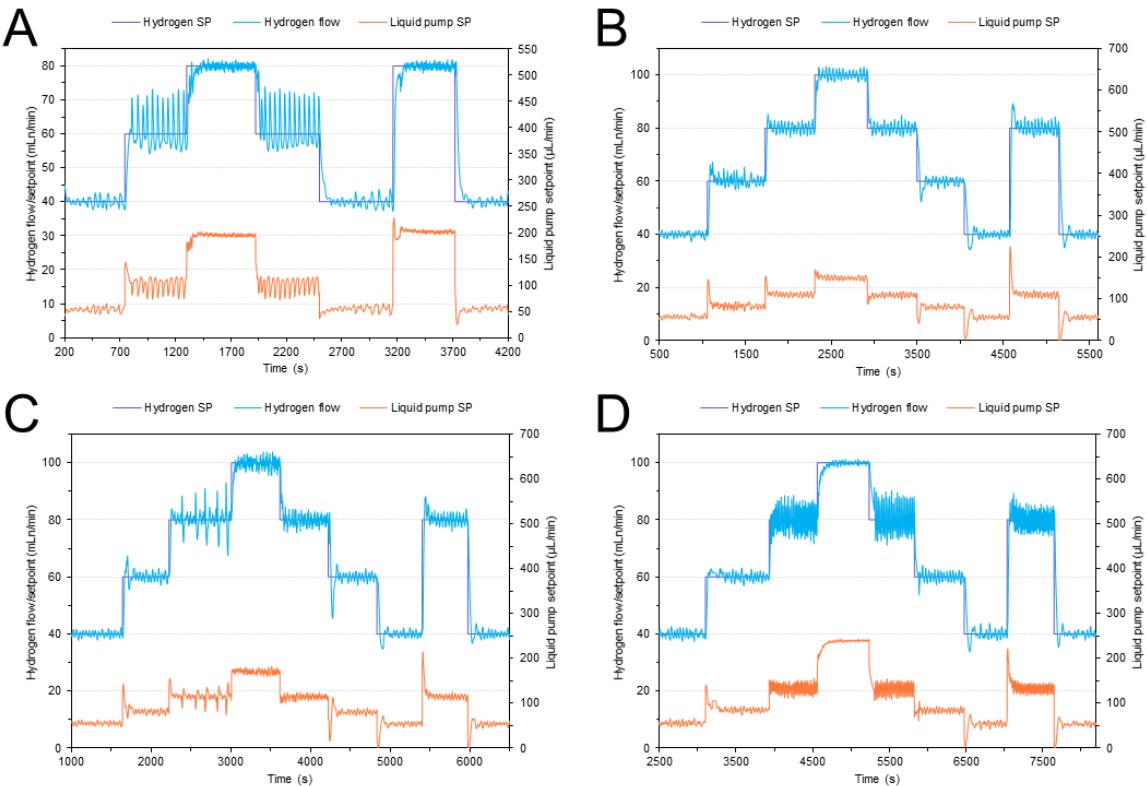

**Figure 6.** Experimental results after implementing the gain-scheduled PI controller in the methanol-reforming plant. Pressure 6 bars (**A**), 8 bars (**B**), 10 bars (**C**), and 12 bars (**D**).

## 4. Conclusions

In this study, a straightforward PI controller with gain scheduling was implemented in a catalytic membrane reactor (CMR) for the production of pure hydrogen via the steam reforming of methanol (S/C = 1). Dynamic experiments were conducted on a laboratory-scale pilot plant to investigate the behaviour of the system. The setpoint for the fuel reformer was defined as the permeation rate of pure hydrogen through the CMR membranes, with the control variable being the liquid flow rate of the methanol and water mixture at the CMR inlet. A comprehensive model was developed, and PI controller parameters, as well as transfer functions, were established for the CMR operating at a temperature of 450 °C and pressures ranging from 6 to 12 bar, reflecting real operating conditions. The developed PI controller effectively maintains the setpoint for the pure hydrogen flow rate under varying operating conditions, offering a commendable response time and remarkable robustness. This represents a valuable strategy for CMR control.

**Author Contributions:** Conceptualization, J.L. and R.T.; methodology, A.C.; formal analysis, A.C., M.S., J.L. and R.T.; writing—review and editing, A.C., M.S., J.L. and R.T.; supervision, J.L. and R.T. All authors have read and agreed to the published version of the manuscript.

**Funding:** This research has been funded by projects MICINN/FEDER PID2021-124572OB-C31, PID2021-126001OB-C31 and GC 2021 SGR 01061.

**Data Availability Statement:** The data that support the findings of this study are available from the corresponding authors upon reasonable request.

**Acknowledgments:** A.C. is grateful to Generalitat de Catalunya and Addlink Software Científico S.L. for Industrial Doctorate grant 063/2018. J.L. is a Serra Húnter Fellow and is grateful to ICREA Academia program. We are grateful to Vicente Roda, Alejandro Martinez, and Isabel Serrano (UPC-EEBE) for technical assistance.

**Conflicts of Interest:** The authors declare no conflict of interest.

## Nomenclature

| | |
|---|---|
| PI | proportional integral controller |
| PID | proportional integral derivate controller |
| CMR | catalytic membrane reactor |
| MSR | methanol steam reforming |
| WGS | water–gas shift reaction |
| S/C | steam to carbon |
| $K_c$ | proportional constant of PID controller |
| $\tau_i$ | integrative constant of PID controller |
| $k_p$ | proportional constant of transfer function |
| $\tau_p$ | integrative constant of transfer function |
| $T_s$ | time delay constant of transfer function |
| SP | setpoint |
| P | pressure, Pa |
| T | temperature, K |
| $CH_3OH$ | methanol |
| CO | carbon monoxide |
| $CO_2$ | carbon dioxide |
| $H_2O$ | water |
| Superscripts | |
| ° | standard condition |
| Subscripts | |
| liq | liquid |

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
