# Peer review of "Experimental Control of a Methanol Catalytic Membrane Reformer"

_reactions, doi:10.3390/reactions4040040_

Round 1

Reviewer 1 Report

Comments and Suggestions for Authors

Reviewing this paper was made more difficult because the references to tables and figures throughout were missing and replaced by Error! Reference source not found and also Figure 2 was inserted twice, possibly obscuring part of the text on page 5.

Apart from the problems mentioned above, this is a short, compact description of a study of a reacting system in a membrane reactor with proportional control of the liquid feed rate to maintain the desired hydrogen flow rate from the permeate side of the membrane. It is mainly experimental with some transfer function modeling.

Although the scope is limited and the study is short, it is still necessary for the authors to state clearly what the original contribution of the work is. They review similar studies for methane steam reforming in the Introduction, is a study for methanol with a hydrogen permeation membrane novel? Or is it the control aspect? It should be made clear.

With attention to these points the work is publishable.

Reviewer 2 Report

Comments and Suggestions for Authors

it could be of interest to understand if the purity of H2 was affected by the the proposed control strategy. In other words, to have an indication of CO into the permeate at different operating conditions. Always related to this, how operational time of the reactor could affect H2 permeation and purity.

But this could be outside the scope of the article.

Reviewer 3 Report

Comments and Suggestions for Authors

The paper is a work on control of membrane reactors for hydrogen production.

The methanol reforming in membrane reactor has been already studied by many authors and the referencing should be improved. From the literature review in the paper it seems that only one group works on such a system.

More details should be given in the experimental part. what about temperature fluctuations due to change in feed flow rates? 

The authors state that the recovery of hydrogen changes with pressure (correct). But it also changes with flow rate. The amount of membranes is fixed, thus the ration between permeation and reaction changes. this should be also discussed.

One would expect that pressure can be used as controlled variable much better than flow rate. The flowrate would impact strongly on the auxiliaries of the system, which is not captured in the experimental setup and thus in the model. 

Comments on the Quality of English Language

polishing of the English is required

Round 2

Reviewer 1 Report

Comments and Suggestions for Authors

Authors have responded to my comments on their original version, and it is now satisfactory.

Reviewer 3 Report

Comments and Suggestions for Authors

Paper corrected and answers are satisfactory. you can accept

Comments on the Quality of English Language

in the proofs stage, check the english again